# IVDR: Analysis of the Social, Economic, and Practical Consequences of the Application of an Ordinance of the In Vitro Diagnostic Ordinance in Switzerland

**DOI:** 10.3390/diagnostics13182910

**Published:** 2023-09-11

**Authors:** Alix T. Coste, Adrian Egli, Jacques Schrenzel, Beatrice Nickel, Andrea Zbinden, Reto Lienhard, Alexis Dumoulin, Martin Risch, Gilbert Greub

**Affiliations:** 1Institute of Microbiology, University of Lausanne & University Hospital Center, 1011 Lausanne, Switzerland; alix.coste@chuv.ch; 2Institute of Medical Microbiology, University of Zurich, 8006 Zurich, Switzerland; aegli@imm.uzh.ch (A.E.); azbinden@imm.uzh.ch (A.Z.); 3Bacteriology Laboratory, Service of Laboratory Medicine, Department of Diagnostic, Geneva University Hospitals, 1211 Geneva, Switzerland; jacques.schrenzel@hcuge.ch; 4Swiss Tropical and Public Health Institute, 4123 Allschwil, Switzerland; beatrice.nickel@swisstph.ch; 5University of Basel, 4002 Basel, Switzerland; 6ADMed Microbiologie Laboratory, 2300 La Chaux-de-Fonds, Switzerland; reto.lienhard@ne.ch; 7Department of Infectious Diseases, Institut Central des Hôpitaux, Hôpital du Valais, 1950 Sion, Switzerland; alexis.dumoulin@hopitalvs.ch; 8Dr Risch Medical Laboratory, Wuhrstrasse 14, 9490 Vaduz, Switzerland; martin.risch@risch.ch; 9Infectious Disease Service, University of Lausanne & University Hospital Center, 1011 Lausanne, Switzerland

**Keywords:** IVDR, microbiology, consequences, risks

## Abstract

IVDR regulation represents a major challenge for diagnostic microbiology laboratories. IVDR complicates a broad range of aspects and poses a risk given the high diversity of pathogens (including rare but highly virulent microbes) and the large variety of samples submitted for analysis. The regular emergence of new pathogens (including Echovirus E-11, Adenovirus 41, Monkeypox virus, Alongshan virus, and Enterovirus D68, as recent examples in Europe in the post SARS-CoV-2 era) is another factor that makes IVDR regulation risky, because its detrimental effect on production of in-house tests will negatively impact knowledge and expertise in the development of new diagnostic tests. Moreover, such regulations negatively impact the availability of diagnostic tests, especially for neglected pathogens, and has a detrimental effect on the overall costs of the tests. The increased regulatory burden of IVDR may thereby pose an important risk for public health. Taken together, it will have a negative impact on the financial balance of diagnostic microbiology laboratories (especially small ones). The already-high standards of quality management of all ISO-accredited and Swissmedic-authorized laboratories render IVDR law of little value, at least in Switzerland, while tremendously increasing the regulatory burden and associated costs. Eventually, patients will need to pay for diagnostic assays outside of the framework of their insurance in order to obtain a proper diagnostic assessment, which may result in social inequity. Thus, based on the risk assessment outlined above, the coordinated commission for clinical microbiology proposes adjusting the IvDO ordinance by (i) introducing an obligation to be ISO 15189 accredited and (ii) not implementing the IvDO 2028 milestone.

## 1. Introduction

On 5 April 2017, the European Parliament approved the new Regulation on in Vitro Diagnostic Medical Devices (IVDR Regulation (EU) 2017/746) [1], which replaces the previous regulation known as the IVDD directive. The new regulation aims to introduce stricter conformity assessment procedures and performance studies to ensure the safety and quality of in vitro medical devices (IVDs). The Swiss Federal Council, to harmonize with the European Union (EU), implemented these changes by adopting the new Ordinance on in Vitro Diagnostic Medical Devices (IvDO RS 812.219) [2] on 4 May 2022 in Switzerland (CH) (see Table 1 for correspondence of terms between EU and CH). These new regulations came into force on 26 May 2022, both in Switzerland and in the EU. This new regulation will lead to significant changes and adaptations for medical laboratories. In this article, we will discuss the difficulties for diagnostic laboratories and the associated risks for Switzerland to comply with this new regulation, with a particular focus on microbiology laboratories. We would like (i) to highlight the impact on the routine diagnostic activity and on the quality of the work, (ii) to question the relevance of continuing with the following steps as planned by this regulation, and (iii) highlight the associated risk for the diagnostic capacities and public health in Switzerland.

## 2. What IVDR and IvDO Imply

The new European law codifies the placing on the market of commercial IVDs produced by industrial manufacturers, and in-house IVDs developed by diagnostic laboratories or larger health institutions themselves, as designated in the IVDR (Table 2). Note that for all devices for general laboratory use, the processing and operating instructions in a given laboratory are not subject to this new regulation [3].

As far as industrial manufacturers are concerned, the number of definitions of an IVD has increased to 74, and the text defines more precisely the different types of diagnostic procedures. The notified bodies (Swissmedic in Switzerland) would therefore be entitled to examine a previously unregulated product for CE marking. In addition, the IVDR introduced a new classification system considering the possible impact on the patient. This replaces the previous categories of IVDs. It divides IVDs into four categories, Class A, Class B, Class C, and Class D, depending on the purpose of the device and the risks involved for patients (Table 3). *De facto*, CE markings obtained under the previous regulation are no longer valid. Each product must be re-certified under the new regulation within different schedules according to the IVD class. Therefore, if the IVD was certified before the 26 May 2022 in accordance with the former legislation, notification must now occur before 26 May 2025 for Class D, 26 May 2026 for Class C, and 26 May 2027 for Class A and B, respectively (Figure 1). To alleviate the effects of a lack of update to the MRA (mutual recognition of conformity assessments) with the EU, the Swiss Federal Council adopted mitigating measures. Therefore, notification in Switzerland for IVDs from EU manufacturers is not necessary if an authorized representative person is domiciled in Switzerland (article 44 paragraph 1 IvDO). In addition, Article 2 of the IVDR specifies that post-market surveillance reports should be updated each year.

Concerning in-house IVDs, also known as laboratory-developed tests (LDTs), their definition, use, and obligation of notification are defined in article 5.5 of the IVDR and articles 9 and 10 of the IvDO. Any test is considered to be LDT when developed and validated according to the EN ISO 15189 norm in a health institution, or any CE-marked commercial test used off label, meaning not precisely following the manufacturer’s recommendations. Thus, for example, the use of CE-IVD tests on other sample types and/or another population from the one(s) listed in the certification are considered off label. Similarly, any modification of the interpretation thresholds or of the manufacturer’s instruction for use defines the test as off label (see Table 2). All LDTs or off-label tests will be submitted to the IVDR/IvDO from 26 May 2024 with different notification deadlines according to the IVD class, i.e., 1 July 2024 for class D, 1 January 2025 for classes B and C, and 1 July 2025 for class A, according to article 90 paragraph 3 of the IvDO (Figure 1). In addition, as of 26 May 2028, to continue using their LDTs, healthcare institutions will have to demonstrate that there is no CE-IVD test with equivalent performance (article 82, paragraph 4 IvDO) or to justify the application of the LDT (Figure 1). In addition, this will have to be demonstrated regularly. This concerns tests that are assumed to be equivalent, such as the use of a home-made database versus a manufacturer’s database on a MALDI-ToF. In no case is this required if the technique used is technically superior to another, typically in cases such as the sequencing of strains versus identification using various biochemical tests.

## 3. Difficulties Encountered in Clinical Microbiology Laboratories

This extremely restrictive law will thus have a profound impact on diagnostic laboratories. In microbiology, this regulation may lead to an overall loss of quality and to a decrease in the portfolio of analyses offered to clinicians, in complete opposition to the objectives of the law (Table 4).

Concerning manufactured IVDs, even before the implementation of the IVDR/IvDO, the availability of diagnostic tests for pathogens that are infrequent or difficult to culture (*Coxiella*, *Bartonella*, *Brucella*, *Rickettsia*, *Chlamydia psittaci*, dengue virus, *Trypanosoma cruzi,* (chagas disease), and many other neglected diseases, was very limited. In addition, some companies were already beginning to discontinue tests in 2022 due to the potential regulatory burden and associated costs, prompting users to urgently evaluate the few remaining tests on the market. In particular, not every company will have a representative in Switzerland. In addition, in the short term, small companies not able to cope with the cost and workload linked to IVDR will have to stop their activity. We will rapidly run into a massive selection of companies with narrow options, which will lead in the medium term to a massive reduction in industrial innovation due to a lack of incentive for smaller start-up companies. 

For example, the main manufacturer of the immunofluorescence tests for the detection of *Coxiella* as well as *Bartonella* stopped production in 2021, leaving only two other commercial alternatives for the Swiss market. The same holds true for the Wright test used for the detection of brucellosis, which is still considered the gold standard. Thus, the current law, despite the adjustments already made to mitigate its risk, and the postponements of its implementation (REGULATION (EU) 2022/112), entails an increased risk of test shortages, in particular concerning infrequent or difficult-to-diagnose pathogens. 

This law adds a critical layer of difficulty for Swiss diagnostic laboratories. Currently, laboratories are already struggling with the massive cost pressure due to the FOPH-induced reduction of 10% and the current explosion in the cost of reagents around the globe; validating cheap in-house assays or switching to more expensive commercial solutions would add additional cost pressure. This additional layer of IVDR regulation is unlikely to improve quality, since the vast majority of microbiology laboratories are ISO 15189 accredited and consequently perform adequate quality control and have implemented a global quality management system. In addition, all microbiology laboratories need to work according to the epidemiology law (Lep—Loi sur les épidémies), which stipulate the necessity of having Swissmedic authorization to practice; the latter is provided for a five-year period. Both aspects are regularly controlled and supervised by both authorities. 

Concerning LDTs, microbiology is a field that must remain on constant diagnostic surveillance in order to be able to react quickly in case of (i) the emergence of new pathogens, such as SARS-CoV-2 virus [7] or highly resistant *Candida auris* [8], (ii) the emergence of new variants of already-known pathogens that could escape the commonly used tests, such as the Swedish variant of *Chlamydia trachomatis* [9], or a *Neisseria meningitidis* variant [10], or (iii) the new epidemiological spread of organisms previously restricted to a geographical area, such as Zikavirus, the Monkeypox virus, and others [11,12,13]. Additionally, when a novel emergent agent is discovered (e.g., the Alongshan virus [14,15]), it is necessary to rapidly establish diagnostic tools for assessing its pathogenic role and dissemination. Thus, the current knowledge and flexibility in implementing new diagnostic tests in Swiss diagnostic laboratories is pivotal. 

In addition, tests are often validated on patients and samples reflecting the more frequent clinical presentations of the disease, but this list cannot be exhaustive. Laboratories are often confronted with the need to use those tests on different patient populations or various sample types. For example, the GeneXpert^®^ Xpert MTB/RIF Ultra test was developed to detect *Mycobacterium tuberculosis* in sputum, but not in biopsies. Similarly, the TPHA and RPR serologic tests for diagnosing syphilis were developed to assess antibodies in serum, plasma but not in the cerebrospinal fluid (CSF) (which is of paramount importance when neurosyphilis is suspected), and many more examples are common in daily practice. 

In all such situations, diagnostic laboratories, and academic laboratories in particular, are on the front line in addressing the urgent and evolving demand for specific, specialized, or new diagnostic assays. Therefore, these labs need to be able to regularly explore and evaluate new tests or off-label usages of commercial tests and to maintain their expertise in that activity. In academic centers, the case selection of patients with rare infectious diseases or who are critically ill is more pronounced. If tests must be sent to a foreign country due to a lack of IVDR/IvDO approval within Switzerland, this may cause significant time delay and present a substantial risk for the patient and public health. The IVDR/IvDO may affect this and other activities due to its substantial administrative workload. In addition, if the law is not changed by 2028, laboratories will have to regularly prove the superiority of their LDTs with respect to manufactured IVDs, generating additional costs and workload due to this extra monitoring activity. Currently, post-market evaluation is guaranteed by regular internal controls and external quality assessments. 

## 4. Foreseen Impact of the Law

The consequences of implementing IVDR/IvDO in microbiology diagnostic laboratories are numerous and potentially severe.

Taken together, the decreased diversity of commercial tests and the strong pressure against LDTs will increase the demand for the remaining commercially available tests, which could lead to increased prices (due to monopoly), company dependence, and logistical challenges, as well as a shortage of reagents for all laboratories. Such a shortage, and now with no redundancy, is likely to have a dramatic effect on patients, and may translate into a drop in diagnostic quality, or even result in a lack of diagnostic capacity in the country. As about 70% to 80% of medical decisions are based on in vitro laboratory diagnostic tests, reduced test availability or even shortages thereof will dramatically impact patient care and potentially lead to clinical/legal consequences. 

The workload and the charge of laboratories will strongly increase with the need (i) to assess all tests developed in house, despite existing clearance by Swissmedic certification and ISO accreditation, and compare them to CE-marked IVDs, (ii) to validate newly implemented CE tests, (iii) to re-assess all tests now considered off label, e.g., due to using different materials, and (iv) to provide notification of all such measures to Swissmedic, the Swiss notified body.

It can be easily imagined that small laboratories will not survive the additional costs of reagents and controls, not to mention the heavy IVDR-related workload, prompting them to reduce their test portfolio, at the risk of becoming less attractive and competitive in retaining their clients. This will result in overall economic damage with unemployment. A heavy transfer of diagnostic tests will take place towards large centers, e.g., academic centers, which will also face increased costs, higher workloads, and a shortage of clinically useful, but less frequently used assays. Lower quality can be expected due to the non-execution of some assays, delays in reporting results due to shipping to foreign centers, increased administration, and, again, additional costs. 

The work of microbiological experts (FAMH) will be hampered. Indeed, with in-house tests, all test details are available, such as the amplification curve of a real-time or quantitative PCR (qPCR) assay, which is not the case for many commercial tests. Usually, laboratory experts know the exact performances of LDTs (sensitivity, specificity, positive and negative predictive value), and their limits (type of samples, patients, potential cross reactions, etc.). In contrast, commercial tests are often black boxes that do not disclose any details, e.g., the composition of the assay, which negatively affects the interpretation and medical validation of the results. 

From 2028 onwards, the virtual obligation to adopt commercial tests will lead to a drastic reduction in the research and development carried out by diagnostic laboratories and start-up companies, as these will become too cumbersome for less frequent usage in the laboratory. This will result in a reduction in innovation in Switzerland, even in new fields of investigation such as next-generation sequencing (NGS) approaches. Indeed, new applications of next generation sequencing NGS (whole-genome sequencing with genome typing and taxo-genomics, and microbiota, resistome, and virulome analyses [16,17,18,19,20]) have for few years been implemented in routine microbiology laboratories. Given the increasing use of genomics for diagnosis, IVDR regulations may impair this recent development.

This will lead to a considerable loss of skills for all laboratory co-workers. Routine development, with regular research and development (R&D) meetings, is helped by having trained persons to rapidly implement new assays (PCR, serology, antigen and culture-based diagnostics). The value of the FAMH experts may ultimately be called into question, with a foreseeable lack of interest in this discipline resulting in personnel shortages in the medium term.

In the long run, we fear a global impoverishment of our diagnostic capacities in microbiology, a loss of knowledge and competence, and a dramatic impact on the quality of service for patients, which is ironically the absolute opposite of the stated goals of this law. Indeed, the law clearly protects industrial manufacturers from the in-house development of devices in the presence of commercial ones, as laboratories will not invest too much money and time into new devices if the ability to use them in future is not protected. The only big winners will be—in the short term—large manufacturers, who will be able to meet the resulting increases in costs and workload, while being given the opportunity to consolidate their business. The dependencies on these large diagnostic companies will further increase. They will direct their efforts toward tests that are in high demand, with high foreseen incomes, neglecting niche investigations, which are, however, the heart of modern precision quality medicine and are of high importance for patients with rare infectious diseases.

## 5. How to Face the Problems Raised by the Regulation

As discussed at the ECCMID 2023 meeting in Copenhagen (15 to 18 April 2023), in session SY241, microbiologists wondered whether we should “wait and see” or “do more lobbying towards authorities” with the aim of modifying the IVDR/IvDO.

The Swiss example of an abrupt cut in test reimbursement by 10% by the FOPH serves as a good indicator that the strategy of “wait and see” is not the way to go. Laboratories already knew that PCR was reimbursed at a rate that was too high, but did not actively alert the Swiss authorities. Millions of francs earned by laboratories during the COVID crisis highlighted the large discrepancy between the real cost of high-throughput PCRs (including reagents, automated systems, and manpower costs) and the level of reimbursement. The discrepancy was especially large given the large number of tests executed each day, creating a situation in which performing a few tests per week for rare pathogens (e.g., *Coxiella burnetii, Chlamydia psittaci,* parasites, exotic viruses such as Monkeypox, West Nile fever virus, Zikavirus, and many more) was not cost-effective, and cross-financing with other more cost-efficient tests was important for running a lab. This sudden cut in test reimbursement had a major impact on the smallest entities (laboratories and hospitals) and laboratoriess specialized in rare diseases, risking a further reduction in test availability in the most remote areas of Switzerland, accelerating imbalances and inequity in our country. 

Conversely, increased lobbying toward authorities seems fruitful. Indeed, the lobbying already undertaken at the European level helped much, achieving (i) increased flexibility (more options to interpret the law according in specific situations), (ii) extended timelines for implementation (Regulation (EU) 2022/112), and (iii) abandonment of the sell-off deadline for manufactured products (Regulation (EU) 2023/607). Lobbying at the Swiss level was also successful in obtaining an attenuation measure by the Swiss Federal council (CE-marked IVD do not need to be accredited by Swissmedic, as long as there is a representative of the company domiciled in Switzerland). However, due to the market size of Switzerland, especially for rare pathogens, this is unlikely to happen. 

Consequently, the Coordinated Commission of Clinical Microbiology of the Swiss Society of Microbiology considers it very important to make all efforts to lobby against this complexification of the regulations, which will clearly fail to meet its objectives. The authors clearly state that the ISO 15189 accreditation (or equivalent accreditation) norm and its routine application, as well as the Swissmedic certification, provide a legal framework that is strict enough to push all laboratories towards high-quality practices, as already exemplified by other medical laboratory disciplines [21,22]. 

To avoid further complexification of current regulations, we recommend adjusting the IvDO regulation as follows:ISO-15189-accredited laboratories may replace all LDT notifications with submission of their accreditation certificate.The IvDO 2028 milestone, which requires the superiority of an LDT against any CE-IVD test to be proven, should not be implemented. Mitigation measures could, for instance, include scientific surveillance and comparison with commercial tests, up to the level of equivalence (but not superiority).

## 6. Conclusions

The IVDR and the IvDO are two examples of bureaucratic decisions that are disconnected from reality and driven by strong financial lobbies. CE-marked assays such as lateral flow have for a long time been shown to often be insufficiently accurate, with low sensitivity (ranging from 30 to 80%), thus clearly calling into question the real value of CE marking, and more generally the competencies of the so-called competent authorities that deliver CE-marking labels. Notably, one of the few possible positive impacts of this new regulation might be a real improvement in CE-labeling procedures, subsequently resulting in increased quality.

However, there is no doubt that IvDO, if left unchanged, will have detrimental effects on patient care and public health by reducing the portfolio of tests; therefore, Swiss clinical microbiologists have decided—by means of this short article—to officially question this new regulation and its application, while proposing some important modifications. We are deeply convinced that laboratories should continue active R&D programs, active training of young microbiologists, and preserve the level of expertise and knowledge in order to continue to serve the aims of patient care, public health, and adaptation to the emerging biological (rather than administrative) threats.

## Figures and Tables

**Figure 1 diagnostics-13-02910-f001:**
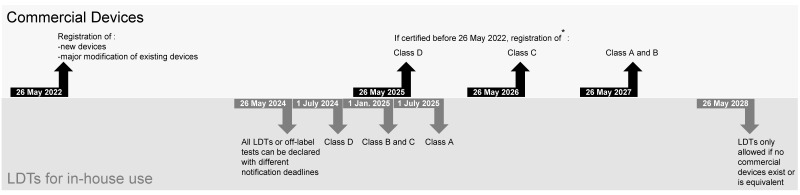
Notification and registration timelines. * Not necessary in Switzerland if already done in EU and authorized representative person domiciled in Switzerland.

**Table 1 diagnostics-13-02910-t001:** Correspondence of terms and entities between the European Union (EU) and Switzerland (CH).

Definition	Designation in EU	Designation in CH
New regulation text and denomination for in vitro diagnostic medical devices	IVDR	IvDO *
Definition of in-house or laboratory-developed tests (LDTs) and obligation to notify	Article 5.5	Articles 9 (definition) and 10 (notification)
Authority to which the notification must be sent	Notified Bodies	Swissmedic
In vitro devices	IVDs	IVDs **
Former regulation	IVDD	MedDO
IVD classes	IVDR Chapter V, Section 1, Article 47	IvDO Section 2, art. 14

* IvDV: die Verordnung über In-vitro-Diagnostika, in German, and ODiv: ordonnance sur les dispositifs médicaux de diagnostics in vitro, in French. ** In french: Dispositifs médicaux de diagnostic in vitro (DIV); in german: IVD (In-vitro-Diagnostika).

**Table 2 diagnostics-13-02910-t002:** Examples of in vitro diagnostics devices affected by the IVD regulations (IvDO/IVDR).

Type of Devices	Targeted	Unaffected
Microscope	Gram examination	Confocal microscopy for research purposes
Thermocyler	In-house diagnostic PCR	Amplification of a gene for cloning purposes (research use only)
Excel file or R pipeline	Calculation of a concentration or parasitemia, comparison of values for test validation	Statistics of ticks Statistical evaluation of results on individual pathogens
DNA extraction kit	DNA extracted from a patient sample for diagnostic purposes	DNA extracted from soil to detect microorganisms
CE-marked devices	- RUO devices used for diagnostic purposes - Commercial IVD devices for which adjustments to the kit protocol were made (modification of cut-off for example) - Commercial IVD devices for which adjustments concerning the intended use (patients, samples, etc.) were made	Commercial IVD devices used following manufacturer recommendations, including intended use and kit protocol

See [4,5] for further details and explanations.

**Table 3 diagnostics-13-02910-t003:** IVD classification definitions (IVDR2017/746 Annex VIII) and examples in microbiology.

Class	Definition	Examples	
D	- Devices intended to be used for the detection of the presence of, or exposure to, a transmissible agent in blood, blood components, cells, tissues or organs, or in any of their derivatives, in order to assess their suitability for transfusion, transplantation or cell administration.	- Hepatitis B (HBs-Ag). - Hepatitis C (Anti-HCV). - Human Immunodeficiency Virus 1/2 (Anti-HIV 1/2).	
	- Devices intended to be used for the detection of the presence of, or exposure to, a transmissible agent that causes a life-threatening disease with a high or suspected high risk of propagation	- Hemorrhagic fever viruses - SARS-CoV and SARS-CoV-2. - MERS Coronavirus. - Smallpox virus.	
	- Devices intended to be used for determining the infectious load of a life-threatening disease where monitoring is critical in the process of patient management	- Hepatitis B Virus (DNA). - Hepatitis C Virus. - Human Immunodeficiency Virus.	
C	- Devices intended for detecting the presence of, or exposure to, a sexually transmitted agent	- *Chlamydia trachomatis.* - *Haemophilus ducreyi.* - Herpes simplex virus 1&2. - Human papilloma virus (HPV). - *Neisseria gonorrhoeae.*	- *Mycoplasma hominis.* - *Mycoplasma genitalium.* - *Trichomonas vaginalis.* - *Treponema pallidum.* - *Ureaplasma urealyticum.*
	- Devices intended for detecting the presence in cerebrospinal fluid or blood of an infectious agent without a high or suspected high risk of propagation	- Bacterial pathogens: *Streptococcus pneumoniae*,* Group B Streptococcus*,* Neisseria meningitidis*,* Haemophilus influenza type B*,* Listeria *spp.,* Borrelia burgdorferi*,* Mycobacterium tuberculosis*. - Fungal pathogens: *Cryptococcus neoformans*,* Aspergillus* spp. - Viral pathogens: Herpes simplex virus 1&2, human herpes virus 6, varicella zoster virus, enterovirus, West Nile virus, chikungunya, Dengue, Zika, hepatitis A, hepatitis E. - Parasitic pathogen: Toxoplasma gondii.	
	- Devices intended for detecting the presence of an infectious agent, if there is a significant risk that an erroneous result would cause death or severe disability to the individual, fetus or embryo being tested, or to the individual’s offspring	- Bacterial pathogens: *Treponema pallidum*,* Chlamydia trachomatis*,* Haemophilus influenzae type B meningitis*,* Neisseria meningitidis*,* Listeria meningitis (Listeria monocytogenes)*,* Mycobacterium leprae, Mycobacterium *spp.,* Legionella *spp.,* Streptococcus agalactiae*, methicillin-resistant *Staphylococcus aureus* (MRSA) and multi-resistant Enterobacteriaceae (MRE). - Parasitic pathogens: *Toxoplasma gondii*. - Viral pathogens: Herpes simplex virus 1&2, cytomegalovirus, Rubella, Measles, Poliomyelitis, Parvovirus B19, Zika.	
	- Devices intended for pre-natal screening of women to determine their immune status towards transmissible agents	- Cytomegalovirus. - Rubella virus. - Toxoplasma gondii.	- Varicella zoster virus. - Zika. - Parvovirus B19.
	- Devices intended for determining infective disease status or immune status, where there is a risk that an erroneous result would lead to a patient management decision resulting in a life-threatening situation for the patient or for the patient’s offspring	- *Salmonella typhi* in feces, for the assessment of the carrier-status of patients. - Antibodies from lymphocyte secretions immunoassay intended for the detection of active *Mycobacterium tuberculosis* infection. - Quantitative virus-specific NAT tests (e.g., Cytomegalovirus, John Cunningham virus, Adenovirus, Enterovirus) to monitor an immunocompromised patient’s (e.g., transplant patient) response to antiviral therapy. - Methicillin-resistant *Staphylococcus aureus* and *Staphylococcus aureus* specific polymerase chain reaction assay for pre-surgical screening of patients to determine nasal carriage. - Assays intended for the detection of IgM antibodies against rubella virus to identify an acute infection in pregnant women to determine whether specific treatment is necessary for protecting the fetus from virus-induced damage due to a lack of previously acquired immunity. - Assays intended for the detection of IgM antibodies against HEV. - Enzyme immunoassay intended for the quantitation of intrathecal antibodies against rubella virus in the diagnosis of rubella virus-induced encephalitis. - Assays intended for the detection of antibodies in the recipient to potentially pathogenic viruses (e.g., anti-cytomegalovirus, anti-herpes simplex virus antibodies) to determine latent disease status of viral infection prior to organ or bone marrow transplantation.	
B	Devices not covered by the other-mentioned classification rules are classified as class B	- Device intended for the detection of *Candida albicans*. - Device intended for the detection of or exposure to *Entamoeba histolytica*. - Test to detect *Helicobacter pylori*, *Clostridioides difficile*, adenovirus, rotavirus, and *Giardia lamblia*. - Biochemical test for establishing the identification of microbiological culture isolates or for determining antimicrobial susceptibility of microbiological culture isolates except those permitting identification or determination of MIC associated with a life-threatening condition. - Antibody tests for HAV, dengue, chikungunya, and West Nile virus. - Assay intended for the detection of IgG antibodies against HEV. - Device intended for the detection of Influenza A/B virus (non-pandemic).	
A	- Products for general laboratory use, accessories which possess no critical characteristics, buffer solutions, washing solutions, and general culture media and histological stains, intended by the manufacturer to make them suitable for in vitro diagnostic procedures relating to a specific examination	- General microbiological culture media containing selecting agents, antimicrobial chromogenic agents, chemical indicators for color differentiation. - Solutions like cleaners, buffer solutions, lysing solutions, diluents specified for use with an IVD. - General staining reagents - Kits for Isolation and purification of nucleic acids from human specimens. - Library Prep reagents for preparation of DNA for downstream analysis by NGS sequencing. - General reagents (not assay specific) used with a Class A instrument, e.g., general sequencing consumable reagents used with a sequencer.	
	Instruments intended by the manufacturer specifically to be used for in vitro diagnostic procedures	Enzyme immunoassay analyzer, PCR thermocycler, sequencer for NGS applications, clinical chemistry analyzer. - Instrument for automated purification of nucleic acids and PCR set-up.	

Table adapted from: Guidance on Classification Rules for in Vitro Diagnostic Medical Devices under Regulation (EU) 2017/746 [6].

**Table 4 diagnostics-13-02910-t004:** Major potential risks for microbiology laboratories due to IVDR.

Risk	Reasons Foreseen
Costs	Increased workload; additional reagents and control costs
Reduction of quality	Less test diversity, fewer laboratories, less innovation, reduced capacity to rapidly react to emerging microbes, decreased skills of co-workers
Lack of tests	Decreased diversity of commercial tests and strong pressure against LDTs will increase the demand for the remaining commercially available tests
Loss of specialists	The regulatory burden and reduced scope for innovation may make the profession seem unattractive.

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
