# Peer review of "IVDR: Analysis of the Social, Economic, and Practical Consequences of the Application of an Ordinance of the In Vitro Diagnostic Ordinance in Switzerland"

_diagnostics, 2023, doi:10.3390/diagnostics13182910_

Round 1
Reviewer 1 Report
This is a timely opinion. It points to dire consequences for patients because of market interruptions. The paper can be improved (gain in strength) by
1. Distinguish between manufacture of devices (subject of the IVDR) and interpretation protocols (medical activity not regulated by IVDR)
2. State that when an in-house approach is selected e.g. because of superiority of a technique, e.g. mass-spec or genetic testing over immunometric methods, follow-up of the market for the inferior technique no longer needed.
3. State that where the investment of industrial partners is protected by the exclusion of in-house devices in the presence of commercial devices, the ad-interim investment of laboratories is not protected
4. Refer to papers coming to similar conclusions in other clinical laboratory fields
Author Response
Reviewer 1:
Comments and Suggestions for Authors
This is a timely opinion. It points to dire consequences for patients because of market interruptions. The paper can be improved (gain in strength) by
- Distinguish between manufacture of devices (subject of the IVDR) and interpretation protocols (medical activity not regulated by IVDR)
Answer: Thank you for this suggestion. We introduce this precision page 3.
“Note that all the devices for general laboratory use, process or operating instructions in a given laboratory are not subject to this new regulation”
- State that when an in-house approach is selected e.g. because of superiority of a technique, e.g. mass-spec or genetic testing over immunometric methods, follow-up of the market for the inferior technique no longer needed.
Answer: Thank you for this suggestion. We introduce this precision page 4.
“This concerns tests that are assumed to be equivalent, such as the use of a home-made database versus a manufacturer's database on a MALDI-ToF. In no case is this required if the technique used is technically superior to another, typically such as sequencing of strains versus identification by various biochemical tests.”
- State that where the investment of industrial partners is protected by the exclusion of in-house devices in the presence of commercial devices, the ad-interim investment of laboratories is not protected
Answer: Thank you for this suggestion. We introduce this precision page 6.
“Indeed, this law clearly protect industrial manufacturers from the developments of in-house devices in the presence of commercial ones, as laboratories will not invest too much money and time for new devices if not protected to be able to use them in the future.”
- Refer to papers coming to similar conclusions in other clinical laboratory fields
Answer : Thank you we add two references, one from chemistry and one from hematology groups, page 7.
“as already highlighted by other medical laboratory disciplines [16, 17].”
--------------------------------------------------
Reviewer 2 Report
Thanks for the opportunity to review the manuscript titled " IVDR: applications of the In Vitro Diagnostic Ordinance in 2 Switzerland and possible impacts". Extremely accurate and thorough analysis revealing the limitation in creativity and work of diagnostic laboratories and glaring protectionism towards manufacturers of diagnostics. The problem affects not only Switzerland, but all European countries and those who want to trade with them, i.e. "opinion" is of international significance. The authors also analyze the diagnostics market related to the availability and access to commercial diagnostic kits and the economic consequences of changes in legislation.
The authors show the meaning of such analyzes in all spheres related to European legislation and their promotion. It would be nice if such analyzes were also conducted in the EC - there are none or they are not in an objective context or they are not given the necessary attention.
The problems of the new regulation will be clearly seen in the occurrence of epidemic and pandemic situations related to new pathogens or variants of already existing ones. On the contrary, after the SARS-CoV-2 pandemic, more freedom should have been given to the diagnostic laboratories to be able to react quickly and in a timely manner, rather than waiting for the manufacturing corporations to react.
This type of analysis should serve as a corrective to legislative bodies.
I have some recommendations related to technical performance, as follow:
In connection with the above-mentioned qualities of the manuscript, the title should be corrected. The title does not reflect the meaning and purpose of the manuscript (whether it is about Switzerland or the World). The title should be such as: “Analysis of the social, economic and practical consequences of the application of an ordinance....”
Table 3: sort the classification from D/C/A/B to D/C/B/A or A/B/C/D
Author Response
Reviewer 2
Thanks for the opportunity to review the manuscript titled " IVDR: applications of the In Vitro Diagnostic Ordinance in 2 Switzerland and possible impacts". Extremely accurate and thorough analysis revealing the limitation in creativity and work of diagnostic laboratories and glaring protectionism towards manufacturers of diagnostics. The problem affects not only Switzerland, but all European countries and those who want to trade with them, i.e. "opinion" is of international significance. The authors also analyze the diagnostics market related to the availability and access to commercial diagnostic kits and the economic consequences of changes in legislation.
The authors show the meaning of such analyzes in all spheres related to European legislation and their promotion. It would be nice if such analyzes were also conducted in the EC - there are none or they are not in an objective context or they are not given the necessary attention.
The problems of the new regulation will be clearly seen in the occurrence of epidemic and pandemic situations related to new pathogens or variants of already existing ones. On the contrary, after the SARS-CoV-2 pandemic, more freedom should have been given to the diagnostic laboratories to be able to react quickly and in a timely manner, rather than waiting for the manufacturing corporations to react.
This type of analysis should serve as a corrective to legislative bodies.
I have some recommendations related to technical performance, as follow:
In connection with the above-mentioned qualities of the manuscript, the title should be corrected. The title does not reflect the meaning and purpose of the manuscript (whether it is about Switzerland or the World). The title should be such as: “Analysis of the social, economic and practical consequences of the application of an ordinance....”
Answer: Thank for this suggestion. We thus modify the title in :
“IVDR: Analysis of the social, economic, and practical consequences of the application of an ordinance of the In Vitro Diagnostic Ordinance in Switzerland”
Table 3: sort the classification from D/C/A/B to D/C/B/A or A/B/C/D
The table 3 was modify with the D/C/B/A format.